

# Exploring New Topography-based Subgrid Spatial Structures for Land Surface Modeling

Teklu K. Tesfa and Lai-Yung (Ruby) Leung

Pacific Northwest National Laboratory, Richland, WA 99352, United States of America

*Correspondence to*: Teklu K. Tesfa (teklu.tesfa@pnnl.gov)



**Abstract**

Topography exerts a major control on land surface processes through its influence on atmospheric forcing, soil and vegetation properties, river network topology and drainage area. Land surface models with spatial structure that captures the spatial

heterogeneity influenced by topography may improve representation of land surface processes. Previous studies found that land surface modeling using subbasins instead of structured grids as computational units improves scalability of simulated runoff and streamflow processes. In this study, new land surface spatial structures are explored by further dividing subbasins into subgrid structures based on topographic properties including surface elevation, slope and aspect. Two methods (Local and Global) of watershed discretization are applied to derive two types of subgrid structures (geo-located and non-geo-located) over the

topographically diverse Columbia River basin in the Northwestern United States. In the Global method, a fixed elevation classification scheme is used to discretize subbasins. The local method utilizes concepts of hypsometric analysis to discretize each subbasin using different elevation ranges that also naturally accounts for slope variations. The relative merits of the two methods and subgrid structures are investigated for their capability to capture topographic heterogeneity and their implications on representations of atmospheric forcing and land cover spatial patterns. Results highlight the relative advantages of the Local

method over the Global method. Comparison between the two types of subgrid structures showed that the non-geo-located subgrid structures are more consistent across different area threshold values than the geo-located subgrid structures. Overall the Local method and non-geo-located subgrid structures effectively and robustly capture topographic, climatic, and vegetation variability important for land surface modeling.



## 1   Introduction

Topography exerts a major control on land surface processes through its influence on atmospheric forcing, soil and vegetation properties, and river network topology and drainage area. Consequently, accurate climate and land surface simulations in mountainous regions cannot be achieved without considering the effects of topographic heterogeneity (Leung and Ghan, 1998; 1995; Ghan et al. 2006). Mountain water resources are particularly sensitive to global warming (e.g., Leung and Ghan 1999; Ghan and Shippert 2006; Mote et al. 2007; Kapnick and Hall 2012). The amplified warming at high elevation due to the lapse-rate effect and snow albedo feedback has large impacts on snowpack accumulation and melt, with consequential effects on runoff and water supply (Leung et al. 2004; McCabe and Clark 2005; Rasmussen et al. 2011).

Topography has dominant control on the spatial pattern of atmospheric forcing including surface temperature, precipitation, incoming and reflected solar radiation. Regions characterized by heterogeneous topography generally exhibit diverse hydroclimatic conditions. For example, stable moisture-rich air lifted by the mountains can produce orographic precipitation that dominates the spatial distribution of cold season precipitation in the western United States (Leung et al. 2003). In mid and high latitude regions topography also influences the partitioning of precipitation into snow and rainfall. In addition, incoming and reflected solar radiation is highly dependent on the orientation of landscapes, which can also have significant impacts on surface hydrology through the effects of radiation on cloud, precipitation, and snow processes (Lee et al. 2015).

Topography is also one of the factors of soil formation, exerting dominant control on the spatial patterns of soil properties over watersheds, e.g., soil depth (Tesfa et al., 2009 and references there in). Soils are generally deeper and finer in texture over valleys compared to the shallower and coarser texture over ridges of watersheds. Through its influence on direct and diffuse solar radiation and consequent effects on soil moisture and evapotranspiration, topography affects spatial pattern of vegetation on a landscape. Different vegetation types grow on different parts of a landscape depending on their water demand and resistance to water stress. In semiarid regions, vegetation types that have high water demand or less resistant to moisture stress grow near streams, while vegetation types resistant to moisture stress can grow further from streams (Tesfa et al., 2011). Topography also determines topology of river network and drainage area, which in turn control surface and subsurface flows (Beven, 1997; Chen and Kumar, 2001). Overall, catchment ecohydrology is strongly influenced by the topography-mediated interactions among vegetation, soil, and river network (Thompson et al. 2011).

Improving representations of land-atmosphere and surface-subsurface interactions influenced by fine scale topography and vegetation has been identified as a grand challenge, motivating the need for hyper-resolution land surface modeling (Wood et al., 2011). While hyper-resolution modeling approaches are being tested at regional (Singh et al. 2015) and continental scales (Maxwell et al. 2015), improving the spatial structures of land surface models to capture the effects of topographic heterogeneity could be crucial to advancing modeling of land-atmosphere interactions in earth system models. Tesfa et al. (2014a; 2014b) demonstrated improved scalability of simulated runoff and streamflow processes when subbasins instead of structured grids are used as computational units in the Community Land Model. The improvements of the subbasin-based land surface modeling in scalability come from its important conceptual advantages in capturing atmospheric forcing and runoff generation processes, both strongly influenced by topography that defines the boundaries of the subbasins.

Discretization of the subbasins to capture spatial heterogeneity influenced by topography may further improve the representation of land surface processes. Ke et al. (2013) evaluated several classification methods to account for subgrid variability of surface elevation and vegetation cover for land surface models with structured grids. To the best of our knowledge, development of subgrid structures for the subbasin-based land surface modeling has not been attempted. The purpose of this paper is to explore subgrid structures that capture topographic heterogeneity and its influences on land surface processes for land surface modeling.



Such subgrid spatial structures are expected to provide a more realistic spatial distribution of surface properties and their influence on climatic variability with more reasonable computational requirement compared to discretizing the domain into fine resolution grid-based representations reported in the literature (e.g., Singh et al., 2015).

Motivated by the significant influences of topographic heterogeneity on land surface processes, we explore new topography-based spatial structures by further dividing subbasins into subgrid structures or subgrid units (also hereafter denoted as SU) to

take advantage of the emergent patterns and scaling properties of atmospheric, hydrologic, and vegetation processes in land surface models. For this purpose, two methods (Global and Local) of subbasin discretization are applied to derive two types of SUs (geo-located and non-geo-located) over the topographically diverse regions of the Northwestern United States. In the Global method, the subbasins are discretized into multiple SUs following the surface elevation classification scheme employed in Leung and Ghan (1998; 1995), combined with classifications of topographic slope and aspect. The local method utilizes concepts of

hypsometric analysis (Willgoose and Hancock, 1998; Sinha-Roy, 2002) combined with classification of topographic aspect to discretize each subbasin into multiple SUs. We evaluate the two discretization methods and spatial structures for their capability to capture topographic heterogeneity and their implications on the representation of the spatial patterns of atmospheric forcing and land cover.

The remainder of the paper is organized as follows: Section 2 describes the study area. Development of the new subgrid

structures is discussed in Section 3. The strategy used to evaluate the methods of subbasin discretization and the subgrid structures are discussed in Section 4. Section 5 presents the results and discussion and finally Section 6 closes with conclusions and recommendations.

## 2 Study Area

To investigate the importance of various watershed discretization methods, the Columbia River basin located in the U.S. Pacific

Northwest is used as a case study. Figure 1 shows the topographic patterns, subbasins with average size equivalent to 1/8th degree grid, elevation ranges of the subbasins, and two subbasins representing extreme topographic properties. The Columbia River basin encompasses both mountainous and low-lying regions. Climatically, the mountainous regions are characterized by low temperature and higher precipitation dominated by snowfall, while, the low-lying regions have warmer temperature and lower precipitation mainly in the form of rainfall. The basin encompasses the largest river in the Pacific Northwest region of

North America, and is the fourth largest river in the United States by discharge volume. Water resources in the basin are dominantly controlled by the high precipitation and snow cover in the mountainous areas.

## 3 Development of new SUs for land surface modeling

Two methods of subbasin discretization are implemented to develop land surface subgrid structures that capture the spatial heterogeneity influenced by topography. Both methods are applied to derive two types of subgrid units (SUs): geo-located and

non-geo-located. The following subsections describe the input data, the two discretization methods and the two types of SUs.

### 3.1 Input data

To derive the subgrid units, the study domain is first delineated into subbasins. We utilize the subbasins equivalent to 1/8th degree grids delineated in Tesfa et al. (2014a; 2014b) using ArcSWAT (Soil and Water Assessment Tool; Neitsch et al., 2005) with the 90m Digital Elevation Model (DEM) and the 15-arcsec river networks from the Hydrological data and maps based on

Shuttle Elevation Derivatives (HydroSHEDS) (Lehner et al., 2008). Although DEMs at resolutions of 30m or finer are available





from the United States Geological Survey (USGS), we use DEM from a global database (i.e., HydroSHEDS) because the main goal of this study is to develop subgrid structures for global land surface models and Earth system models. Along with the delineation of the subbasins, topographic attributes such as slope and aspect are derived for the study domain to be used as inputs for the subbasin discretization methods.

### 3.2 Global Method

In the Global method, the study domain is first discretized into 12 elevation classes based on surface elevation extracted from the 90m HydroSHEDS' DEM, following the surface elevation classification scheme employed in Leung and Ghan (1998; 1995) that uses class intervals of 100m for surface elevation below 500m, and gradually increasing to intervals of 500m and 1000m for high surface elevations, resulting in 12 elevation classes (see Figure 2). This method is Global because the same elevation classification scheme is used to discretize all subbasins regardless of the elevation spanned by individual subbasins, which can vary substantially. Since topography influences atmospheric and land surface processes through surface elevation, slope and aspect, the Global method combines topographic slope and aspect with the elevation classes. For this purpose, the study domain is also partitioned into two classes of topographic slope where slope values less than or equal to 20 degrees are grouped as gentle to moderately steep areas, and slope values greater than 20 degrees are grouped as steep to very steep areas following definitions of slope classes by the Natural Resources Conservation Service of the United States Department of Agriculture. Similarly, the study domain is partitioned into two classes of topographic aspect, where areas facing north, northwest, northeast and east are assigned to one class and areas facing south, southwest, west and southeast belong to a separate class. For each subbasin, classes of elevation, slope and aspect are extracted following the subbasin boundary and converted from raster to polygon shapes, resulting in three sets of SUs, respectively, derived based on elevation, slope and aspect separately. The SUs derived from elevation, slope and aspect are then intersected to generate SUs based on the combination of topographic elevation, slope and aspect, resulting in a large number of SUs for each subbasin. Since many of the SUs are extremely small in size, and our goal is to capture topographic heterogeneity with only a reasonable number of SUs for computational efficiency, an area threshold value is used to merge SUs with area smaller than the threshold to their neighboring SUs with size larger than or equal to the threshold value. The Global method has been implemented in Python and utilizes ArcGIS functionalities. In this effort, the Global method is applied to derive both geo-located and non-geo-located SUs.

### 3.3 Local Method

In the Local method of subbasin discretization, the subbasins for the study domain are first classified into five groups based on values of elevation range using the Natural Breaks (Jenks) classification method in ArcGIS. As an example, to derive the hypsometric curves, the two contrasting subbasins shown in Figure 1 are discretized into 100 elevation contours using elevation data extracted from the 90m resolution DEM from HydroSHEDS. The relative elevation (RH) and relative area (RA) are calculated for each contour, where, relative elevation (RH) is defined as the ratio of the height of the given contour (h) from the base plane of the subbasin to the maximum height of the subbasin (H), while relative area (RA) refers to the ratio of the area above a particular contour (a) to the total area of the subbasin (A).

$$RA = \frac{a}{A} \qquad\qquad (1)$$

$$RH = \frac{h}{H} \qquad\qquad (2)$$



The hypsometric curves are derived by plotting the relative area (RA) along the abscissa and the relative elevation (RH) on the ordinate axes. In geomorphology, hypsometric curve is used to characterize the distribution of elevation within a basin. Following Willgoose and Hancock (1998), three parts of the hypsometric curve are identified as the head, body and toe of the subbasin, respectively, and defined as: (1) the upward-concave part of the curve in the upper left-hand side; (2) the downward-

concave part of the curve on the right hand side; and, (3) the upward-concave region in the center of the curve between the head and toe. As shown in Figure 3, relative area values of 0.2 and 0.8 are used to discretize the hypsometric curve into the three parts, following Sinha-Roy (2002). Furthermore, the body part of the subbasins is divided at relative area value of 0.5 for more homogenous topography within each class. As a result, each subbasin is discretized into four elevation bands with elevation class break values at the minimum and maximum elevation, and at relative areas of 0.2, 0.5 and 0.8. Elevation ranges are calculated

between each consecutive class break values. The values of elevation ranges and class break are further used in the algorithms in Tables 1 and 2 to derive the elevation-based SUs for each subbasin. This method is Local as the elevation ranges used to discretize the subbasins vary depending on the topographic variations within each subbasin.

Similar to the Global method, classes of topographic aspect are extracted for each subbasin and intersected with the corresponding elevation classes classifying the subbasin into multiple SUs, where some of them are extremely small in size.

Since discretizing the subbasins using hypsometric curve is expected to capture slope variation implicitly, topographic slope is not used in the Local method. With the main goal to capture topographic heterogeneity with only a reasonable number of SUs for computational efficiency, area threshold is utilized to merge those SUs with area smaller than the threshold to their neighboring SUs with size larger than or equal to the threshold to develop the final SUs. This method has also been implemented in Python and utilizes ArcGIS functionalities. This method is also applied to derive both geo-located and non-geo-located SUs. The actual

number of SUs of each subbasin depends on the variability of surface elevation and topographic aspect within the subbasin boundary.

### 3.4  Types of Subgrid Units

Two types of SUs are derived using both the Global and Local methods: geo-located and non-geo-located. The geo-located SUs are derived by discretizing the subbasins into spatially contiguous structures. They are characterized with explicit geographical

location and a single boundary. In this case, SUs with the same topographic characteristics at different locations of the subbasin are treated as separate units. The non geo-located SUs are developed by discretizing the subbasins into spatially non-contiguous structures. In this case, SUs with the same topographic properties at different locations of the subbasin are treated as a single unit resulting generally in reduced number of SUs compared to the geo-located SUs.

### 4    Evaluation Strategy

### 4.1  Analysis using SUs based only on elevation

Because topographic slope is not explicitly used in the Local method, it is logical to ask whether discretizing subbasins using the hypsometric curve is capable of implicitly capturing the variability of topographic slope within the subbasins. To investigate this, geo-located SUs are derived using both Global and Local methods based on elevation classification only. The number of SUs for each subbasin from both methods is compared against the average values of topographic slope of the subbasins in the study area

to determine how topographic slope influences the number of SUs needed to capture subgrid topographic variability in each method. In addition, the spatial pattern of the number of SUs for each subbasin derived using each method is compared against the spatial pattern of topographic slope and elevation range within the subbasins for the study region. An effective subgrid





method would allow more SUs in subbasins with complex terrain to capture the subgrid topographic variability and use fewer SUs in subbasins with small variations of topography. Finally, the relative capability of the two methods in capturing
topographic heterogeneity and their sensitivity to the values of area threshold are evaluated, respectively, based on the standard deviation of the 90m resolution elevation within the SUs and the variation of statistical metrics (the total number of SUs, mean SU size and standard deviation in SU size) calculated for the study domain across different values of area threshold (1%, 2%, 3%, 4%, & 5%). Methods that are less sensitive to the values of area threshold can provide more robust SUs for representing subgrid topographic heterogeneity.

## 4.2 Analysis using SUs based on elevation, slope, and aspect


The two types of SUs are expected to differ in their ability to capture topographic heterogeneity, the number of SUs, which has important implications to the overall computational burden, and their sensitivity to area threshold values, which is important for defining robust SUs for land surface modeling. Thus, to evaluate the two types of SUs with respect to their applications in land surface modeling, geo-located and non-geo-located SUs for the study area are derived based on elevation, slope and aspect using
both Global and Local discretization methods at different values of area threshold (1%, 2%, 3%, 4%, & 5%). The geo-located and non-geo-located SUs of each method are then compared for their sensitivity across values of area threshold using statistical metrics (total number of SUs, average size of SUs and standard deviation in SU size) calculated over the study domain at different values of area threshold.

The Global and Local methods are further investigated for their capability in capturing topographic heterogeneity and
consistency across different values of area threshold when using the non-geo-located SUs. The relative capability of the non-geo-located SUs from both methods in capturing topographic heterogeneity is evaluated based on the values of standard deviation in surface elevation calculated at each SU across different values of area threshold. In addition, sensitivity of the two methods (Global and Local) when used to derive non-geo-located SUs is evaluated using statistical metrics calculated over the study domain such as total number of SUs, average size of SUs and standard deviation in SU size.

## 4.3 Implications to representation of land surface processes


Since the main goal of this study is to derive land surface structures capable of improving representation of land surface processes in land surface modeling, it is logical to ask how the new structures impact the representation of land surface parameters. For this purpose, the two methods are first evaluated for their relative capability to capture climatic and land cover variability over the study area using the non-geo-located SUs derived at different values of area threshold. The capability to
capture climatic variation is investigated by comparing values of standard deviation in precipitation and surface temperature within the SUs derived using the two methods. In this case, the precipitation and surface temperature datasets for the study area are extracted from the 30 year normal annual precipitation and mean annual surface temperature obtained from the PRISM climate datasets (800m spatial resolution) (http://www.prism.oregonstate.edu/). Similarly, using the Normalized Difference Vegetation Index (NDVI) data as a proxy for land cover, the relative capability of the two methods in capturing land cover
pattern over the study domain is investigated by comparing values of standard deviation in NDVI calculated within the SUs from the two methods. For this purpose, the NDVI datasets for the study area are obtained from the enhanced Moderate Resolution Imaging Spectroradiometer (eMODIS) data (250m spatial resolution) portal (http://earthexplorer.usgs.gov/) at the Earth Observation and Modeling Facility (EOMF). Furthermore, the relative advantages of the non-geo-located SUs derived using the Local method in capturing climatic variability in the study domain are investigated by comparing precipitation and surface



temperature represented using the SUs against those of subbasin-based and original high resolution PRISM grid-based representations.

## 5 Results and discussion

### 5.1 Global versus Local methods using elevation-based SUs

Since the main differences between Global and Local methods are in the way subbasins are discretized into elevation classes and

whether topographic slope is included explicitly, the relative capability of the two methods in capturing topographic heterogeneity is investigated using elevation-based SUs. Figure 4 compares how well the Global and Local subbasin discretization methods capture the topographic slope using elevation-based geo-located SUs derived based on elevation at 1% area threshold. For this purpose, the numbers of SUs per subbasin resulted from both methods are compared against the average topographic slope calculated over the subbasins. The results show the number of SUs per subbasin from the Local method is

directly related to the average subbasin slope (r2 = 0.47), so the steep subbasins are generally discretized into more SUs than the flat subbasins. On the other hand, the number of SUs per subbasin from the Global method is not related (r2 = 0.07) to the average topographic slope of the subbasins. From this comparison, it is clear that discretizing subbasins following the algorithms in Tables 1 and 2 using the hypsometric curve characterization within subbasins is able to capture topographic slope implicitly, making the Local method superior over the Global method.

The Columbia River basin encompasses diverse topography ranging from flat to steep mountainous areas making it an ideal study area for evaluating the relative capability of the two subbasin discretization methods in capturing the spatial pattern of topographic properties. The spatial pattern of the numbers of elevation-based geo-located SUs per subbasin derived using both methods with a 3% area threshold are compared against the spatial pattern of the average topographic slope and elevation ranges of the subbasins classified based on the Natural Breaks (Jenks) classification method in ArcGIS (Figure 5). The results suggest

that the spatial pattern of the number of SUs per subbasin for the SUs from the Local method follows the topographic pattern in the study area better than those of the Global method, confirming further the advantages of discretizing the subbasins using the Local method. The number of SUs per subbasin from the Local method mimics the topographic pattern so more SUs are defined per subbasin over the mountainous areas and fewer SUs are needed per subbasin over the flat areas of the basin. This enables the model to capture the topographic heterogeneity with minimum number of SUs over the study domain, which is essential for

computational efficiency in land surface modeling.

Figure 6 shows the relative capability of the two methods in capturing subgrid topographic heterogeneity across different values of area threshold using elevation-based geo-located SUs. For this purpose, values of standard deviation in elevation within the geo-located SUs derived using different values of area threshold (1%, 2%, 3%, 4%, & 5%) from both methods are compared. The results again clearly show that the SUs from the Local method are able to capture topographic heterogeneity, which is

reflected in the smaller standard deviation of topography within each SU across different values of area threshold, better than those of the Global method. In addition, the results also show that the Local method can capture topographic heterogeneity more consistently across different values of area threshold than the Global method, suggesting that the SUs derived using the Local method are more robust.

Using the same SUs, the two methods are further investigated for their sensitivity to values of area threshold using the variability of statistical metrics (total number of SUs, mean SU size and standard deviation in SU size) calculated over the whole study

domain for different values of area threshold. The results in Figure 7 show that SUs derived using Local method remain more



consistent across different values of area threshold than those of the Global method, making the Local method more robust than the Global method for land surface modeling.

### 5.2 Geo-located versus non geo-located SUs

To evaluate the robustness of the two types of SUs (geo-located and non-geo-located) for land surface modeling, we compare their sensitivity to values of area threshold. For this purpose, geo-located and non-geo-located SUs are derived based on elevation, slope and aspect using both methods at different values of area threshold. The geo-located SUs from each method are then compared against the corresponding non-geo-located SUs derived using the same method based on the statistical metrics calculated over the whole study area. Shown in Figure 8 are comparisons of the variability of the total number of SUs, average

SU size and standard deviation in SU size calculated for the geo-located SUs against those of the non-geo-located SUs for both the Global (Figures 8a, 8b, and 8c) and the Local (Figures 8d, 8e, and 8f) methods. In both methods, the results generally suggest that the non-geo-located SUs are more consistent across different values of area threshold than the corresponding geo-located SUs. Thus, in subsequent sections, the two methods of subbasin discretization are evaluated using the non-geo-located SUs only.

### 5.3 Global versus Local methods using non geo-located SUs

Following the evaluation of the two methods using elevation-based geo-located SUs, it is important to investigate whether the advantages of the Local method over the Global method shown in previous results still apply when the two methods are used to derive non-geo-located SUs based on the combination of multiple topographic properties. Shown in Figure 9 are comparisons of sensitivity of the Global and Local methods to values of area threshold when the two methods are applied to derive non-geo-located SUs using the variability of the statistical metrics (total number of SUs, average SU size and standard deviation in SU

sizes) calculated over the whole study domain at different values of area threshold. Note that unlike the comparison in Figure 7, the SUs in this comparison are non-geo-located, derived based on a combined classification of elevation and topographic slope and aspect in the Global method and elevation and topographic aspect in the Local method. Similar to the comparisons in Figure 7, the results suggest that the SUs from the Local method are less sensitive to the values of area threshold, yielding more consistent values of the total number of SUs, average SU size and standard deviation in SU sizes over the study domain than

those of the Global method.

Shown in Figure 10 are values of standard deviation in elevation within the non-geo-located SUs derived using the Global and Local methods at different values of area threshold, comparing the capability of the two methods in capturing topographic heterogeneity when used for non-geo-located SUs. Similar to the results shown in Figure 6, there is a clear difference in the capability of the two methods in capturing topographic heterogeneity across different values of area threshold. The non-geo-

located SUs from the Local method are able to capture topographic heterogeneity much better than those of the Global method across different values of area threshold. The improved capability of the Local method shown in this comparison comes from the advantage of performing elevation discretization based on hypsometric curve characterization in the Local method (see Figures 6 and 3). The results also suggest that the capability of the non-geo-located SUs from the Local method in capturing topographic heterogeneity remains more consistent at different values of area threshold than those of the Global method, confirming the

superior advantages of the Local method.

From the results shown so far, relative to the Global method, the SUs from the Local method are superior in capturing topographic heterogeneity yielding more SUs per subbasin over mountainous areas and fewer SUs per subbasin over flat areas, which is essential for more realistic representations of the spatial distributions of precipitation and snow cover in mountainous areas and computational efficiency in land surface modeling. Also, the SUs from the Local method are more consistent across



different values of area threshold than those of the Global method. Subsequently, it is important to examine whether similar advantages exist for the Local method in capturing climatic and land cover variability as compared to the Global method. The following section focuses on the implications of the non-geo-located SUs in the representations of climatological and land cover variability in the study area.

### 5.4 Implications to the representation of land surface processes

Topography can influence land surface processes through its impacts on atmospheric forcing and vegetation variability. Consequently, it is essential to examine the implications of the new SUs on representations of climatic and vegetation variability. Shown in Figure 11 are values of standard deviation calculated for the 30 year normal annual precipitation (Figure 11a) and mean annual surface temperature (Figure 11b) obtained from the PRISM dataset within the non-geo-located SUs derived using the Global and Local methods at different values of area threshold, comparing the relative capability of the two methods in

capturing climatic variability in the study area. The results show generally lower values of standard deviation in both precipitation and temperature for the SUs derived using the Local method than those of Global method across all values of area threshold. Consistent with the comparison on the capability to capture topographic heterogeneity shown in Figures 6 and 10, these differences reflect the dominant control of topography and the impact of spatial structure on precipitation and surface temperature, suggesting improved capability in capturing climatic variability for the Local method.

Furthermore, shown in Figure 12 are values of standard deviation of NDVI calculated at the non-geo-located SUs from both Global and Local methods at different values of area threshold. In this comparison, the NDVI is used as a proxy for land cover information during spring (Figure 12a) and summer (Figure 12b) extracted from the eMODIS dataset, showing the relative capability of the two methods in capturing land cover variability in the study area. The results generally show lower values of standard deviation for the SUs derived using the Local method than those of the Global method across all values of area

threshold, suggesting that the SUs from the Local method have better capability of capturing land cover variation in the study domain, which is essential to representation of land cover in land surface modeling.

    In all the results shown so far, the SUs from the Local method have demonstrated clear advantages in capturing topographic heterogeneity and climatic and land cover variation compared to those of the Global method over the study domain. Therefore, we further examined how representation of climatological forcing improves when using the non-geo-located SUs derived using

the Local method at 3% area threshold value compared to the subbasin-based representation. Figure 13 compares the spatial pattern of the 30 year normal precipitation when represented based on subbasins at roughly 1/8th degree resolution (Figure 13a), non-geo-located SUs within the subbasins (Figure 13b) and the original grid representation from the PRISM dataset at 800 m resolution (Figure 13c). Note that the Canadian part of the study domain is missing from the map because the PRISM data are only available for the United States. The results show that the SU-based representation yields similar spatial pattern of

precipitation to that of the original PRISM grids with no visually discernible difference. The spatial pattern of precipitation for the subbasin-based representation has noticeable differences from those of the SUs and original PRISM grid representations. With the ability to better capture the spatial heterogeneity of precipitation, land surface models that use the SU-based representation are expected to produce more realistic distribution of snow cover over the mountains compared to the subbasin-based representation, and it is considerably more efficient computationally compared to modeling land surface processes using

hyper-resolution such as the PRISM grids. Further comparison of the three representations of precipitation using statistical metrics (average and standard deviation values) reveals that the values from the SU-based representation are much closer to those of the original PRISM grids as compared to the subbasin-based representation (Table 3).




Similar comparisons are shown in Figure 14 for surface temperature. Similar to the results for precipitation, there is no visually noticeable difference in the spatial pattern of temperature between the SU-based and original PRISM grid-based representations, while the subbasin-based representation misses important variability indicated in the PRISM data. Comparison using statistical metrics of temperature (Table 3) confirms the advantages of the new SUs representation.

The advantages demonstrated for the SUs derived using the Local method in representing topographic features are expected to be significant for land surface modeling in mountainous areas such as the Columbia River Basin, where topography has dominant control on precipitation and temperature characteristics that translate to differences in runoff and streamflow characteristics (Tesfa et al., 2014a; Tesfa et al., 2014b).

## 6  Summary and Conclusions

Topography exerts a major control on land surface processes through its influence on atmospheric forcing, soil and vegetation properties, network topology and drainage area. Thus, spatial structure of land surface models that captures spatial heterogeneity influenced by topography may improve modeling of terrestrial water cycle and land-atmosphere interactions. In land surface modeling, such spatial structures are very much needed for accurate simulations of land surface processes in Earth system models. In this study, we developed new land surface spatial structures by further discretizing subbasins into subgrid units (SUs) based on their topographic attributes (e.g. elevation, topographic slope and aspect). Two methods of watershed discretization (Local and Global) have been developed and applied over the Columbia River basin in Northwestern United States to derive two types of topography-based subgrid structures (geo-located and non-geo-located). In addition, the two methods have been evaluated for their consistency, capability to capture topographic heterogeneity and climatic and land cover variability of the study domain using both types of subgrid structures.

In the Global method, the study domain is initially discretized into 12 elevation classes following the surface elevation classification scheme employed in Leung and Ghan (1998; 1995). Then, following the subbasin boundary, the elevation classes are intersected with classes of topographic slope and aspect, discretizing each subbasin into multiple subgrid units. The local method utilizes concepts of hypsometric analysis to first discretize each subbasin into elevation classes using algorithms developed in this study, which are then merged with classes of topographic aspect to divide the subbasin into multiple subgrid units. In both methods, values of area threshold are used to merge small subgrid units into the neighboring large subgrid units, yielding reasonable number of subgrid units per subbasin. Both methods are applied to derive two types of subgrid structures: geo-located (spatially contiguous) and non-geo-located (spatially non-contiguous). Furthermore, using both types of SUs, the two methods of subbasin discretization are investigated for their capability to capture topographic heterogeneity, their implications on representations of climatic and vegetation variability in the study area, as well as their sensitivity to the area threshold values.

Using elevation-based geo-located subgrid untis, comparison of the two methods showed that the Local method is able to capture the topographic variability better than the Global method. Taking advantage of hypsometric analysis, the Local method can capture slope variability implicitly so it generally requires fewer SUs to represent subgrid topographic variability. The Local method more effectively captures the topographic pattern across the region by discretizing steep subbasins into more subgrid units and flat subbains into fewer subgrid units. Using the Local method, the standard deviation of surface elevation within the subgrid units is noticeably smaller and less sensitive to the values of area threshold than the Global method. Hence the Local method is clearly more effective and robust for representing subgrid elevation variability for land surface modeling.



Comparing the two types of subgrid structures derived using the Global and Local methods revealed that the non-geo-located SUs are more consistent than the geo-located SUs across different area threshold values. Further investigation of the relative capability of the two methods with non-geo-located subgrid units representing multiple topographic features (elevation, slope, and aspect) based on the standard deviation in surface elevation within the subgrid units and statistical metrics calculated over the whole study domain further demonstrated superior capability and consistency for the Local method compared to the Global

method. Similarly, investigation of the relative capability of the two methods in capturing climatic and land cover variability based on the high resolution PRISM precipitation and surface temperature and NDVI data, respectively, reveals that the Local method is generally better than the Global method. Finally, comparing the precipitation and surface temperature over the study area when represented using non-geo-located SUs from the Local method against those of the subbasin-based and original PRISM grid-based showed the spatial pattern and statistical values of the subgrid units are much closer to those of the original

PRISM grids than those of the subbasins.

In summary, this study demonstrated that adopting the hypsometric curve characterization for discretizing subbasins yields improved capability in capturing topographic heterogeneity and consistency across different values of area threshold. This resulted in improved representation of climatic and land cover variability in land surface modeling. The improved capability to capture subgrid variability of atmospheric forcing, surface topography, and vegetation cover with nominal increase in

computational requirement is essential for improving simulations of land surface modeling in mountainous regions. The focus in this paper is the development and evaluation of the methods and new spatial structures. Future efforts will implement the non-geo-located SUs from the Local method in a land surface model based on the Community Land Model subgrid structure to investigate if or how the addition of topographic subgrid units to the subgrid hierarchical structure translates to improved simulations of evapotranspiration, soil moisture, snowpack, and runoff and streamflow.



**7    Code Availability**

The updated code is available upon request. Please contact Teklu K. Tesfa at teklu.tesfa@pnnl.gov.



**Acknowledgement**

This research was supported by the Office of Science of the U.S. Department of Energy as part of the Earth System Modeling

program. The Pacific Northwest National Laboratory is operated by Battelle for the U.S. Department of Energy under Contract
DE-AC05-76RLO1830.



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



**Table 1**: Algorithm applied to derive elevation-based SUs using the Local method.

**Algorithm 1**: Local method of subbasin discretization. Array $BRK_i = [Elv_{min}, Elv_{0.2}, Elv_{0.5}, Elv_{0.8}, $ and $Elv_{max}]$ denotes elevation values at the initial class breaks, where $Elv_{min}$, $Elv_{0.2}$, $Elv_{0.5}$, $Elv_{0.8}$, and $Elv_{max}$ refer to the minimum elevation, elevation values at relative areas of 0.2, 0.5, and 0.8, and maximum elevation of the subbasin, respectively. Array $R_i = [R_1, R_2, R_3, R_4]$ denotes the values of elevation range between consecutive $BRK_i$. Variables $BRK_f$ denotes the final values of elevation at class breaks. Variable thr denotes the value of elevation threshold (100 m). Variable n denotes the number of Rs with values less than thr. Function GetFinalBRKs() denotes a function used to determine $BRK_f$ by recursively merging Rs less than teh thr with the neighboring Rs recursively.

```
For each Subbasin:
        Derive a hypsometric curve
        Determine elevation values at the BRKi
        Calculate values of Ri between consecutive BRKi
        Determine n

        If n == 0 // All values of Ri greater than the thr
                BRKf = BRKi
                Rf = Ri
        Else if R1 >= thr and R2 < thr and R3 < thr and R4 >= thr:
                If (R2 + R3) >= thr:
                        BRKf = [Elvmin, Elv0.2, Elv0.8, Elvmax] // Keep the body as separate class
                Else:
                        BRKf = [Elvmin, Elv0.5, Elvmax] // Split the body into the head and tail
        Else if R1 >= thr and R2 < thr and R3 >= thr and R4 >= thr:
                BRKf = [Elvmin, Elv0.2, Elv0.8, Elvmax]          // Keep the body as separate class
        Else if R1 >= thr and R2 >= thr and R3 < thr and R4 >= thr:
                BRKf = [Elvmin, Elv0.2, Elv0.8, Elvmax]          // Keep the body as separate class
        Else:
                BRKf = GetFinalBRKs(BRKi, Ri, thr) // Call the recursive function

        Return BRKf
```





**Table 2**: Algorithm to determine the final class break values (BRK$_f$) by merging elevation ranges with less than the threshold to
the neighboring elevation ranges recursively.

---

**Algorithm 2**: To determine the final values of class breaks using recursive function GetFinalBRKs(). BRK$_i$, R$_i$,
n, thr, denote the same variables as in Algorithm 1 (Table 1). Variables i and nn denote an index values of BRKs
and the number of all Rs, respectively.

```
Function GetFinalBRKs(BRKᵢ, Rᵢ, thr):
        Determine n
        Determine nn // number of all Rs
        Determine i // index of Rs with less than thr
        If n > 0 and nn > 1:
                Get the index (i)
                If i == 0: // R is at the beginning of the array
                        Rᵢ[i + 1] = Rᵢ[i + 1] + Rᵢ[i] // merge R with the next neighbor
                        Update BRKᵢ
                        Call GetFinalBRKs(BRKᵢ, Rᵢ, thr) // This is a recursive call
                Else if i == nn: // R is at the end of the array
                        Rᵢ[i - 1] = Rᵢ[i - 1] + Rᵢ[i] //merge R with the previous neighbor
                        Update BRKᵢ
                        Call GetFinalBRKs(BRKᵢ, Rᵢ, thr) //Recursive call
                Else: // merge with the smaller negibor
                        If Rᵢ[i - 1] > Rᵢ[i + 1]
                                Rᵢ[i + 1] = Rᵢ[i + 1] + Rᵢ[i] // merge R with the next neighbor
                                Update BRKᵢ
                                Call GetFinalBRKs(BRKᵢ, Rᵢ, thr) // This is a recursive call
                        Else:
                                Rᵢ[i - 1] = Rᵢ[i - 1] + Rᵢ[i] // merge R with the previous neighbor
                                Update BRKᵢ
                                Call GetFinalBRKs(BRKᵢ, Rᵢ, thr) // This is a recursive call

        Return BRKᵢ
```



**Table 3**: Comparing the SU and Subbasin representations against the original PRISM grid representation using statistical summary of precipitation and surface temperature calculated over the study domain

| Representation | Precipitation (mm) | | Temperature (C°) | |
|---|---|---|---|---|
| | Average | Standard deviation | Average | Standard deviation |
| Subbasin | 669.036 | 459.479 | 7.179 | 2.525 |
| Subgrid Units | 739.051 | 506.828 | 6.782 | 2.664 |
| Original PRISM Grid | 717.021 | 519.523 | 6.935 | 2.681 |


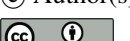



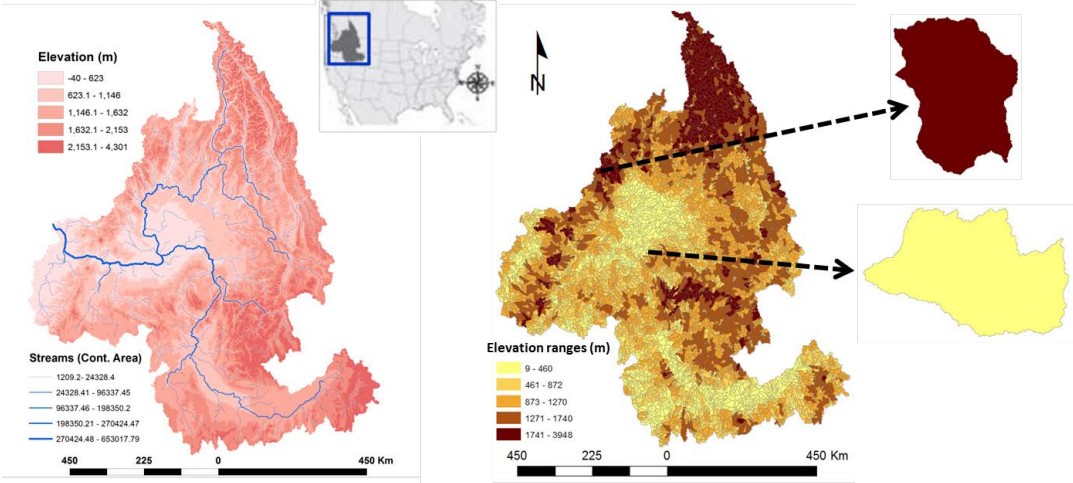

**Figure 1:** The topographic distribution (left) and subbasin delineation (right) of the study area (Columbia River Basin). Two subbasins selected to represent the extreme classes of elevation ranges are shown on the far right.




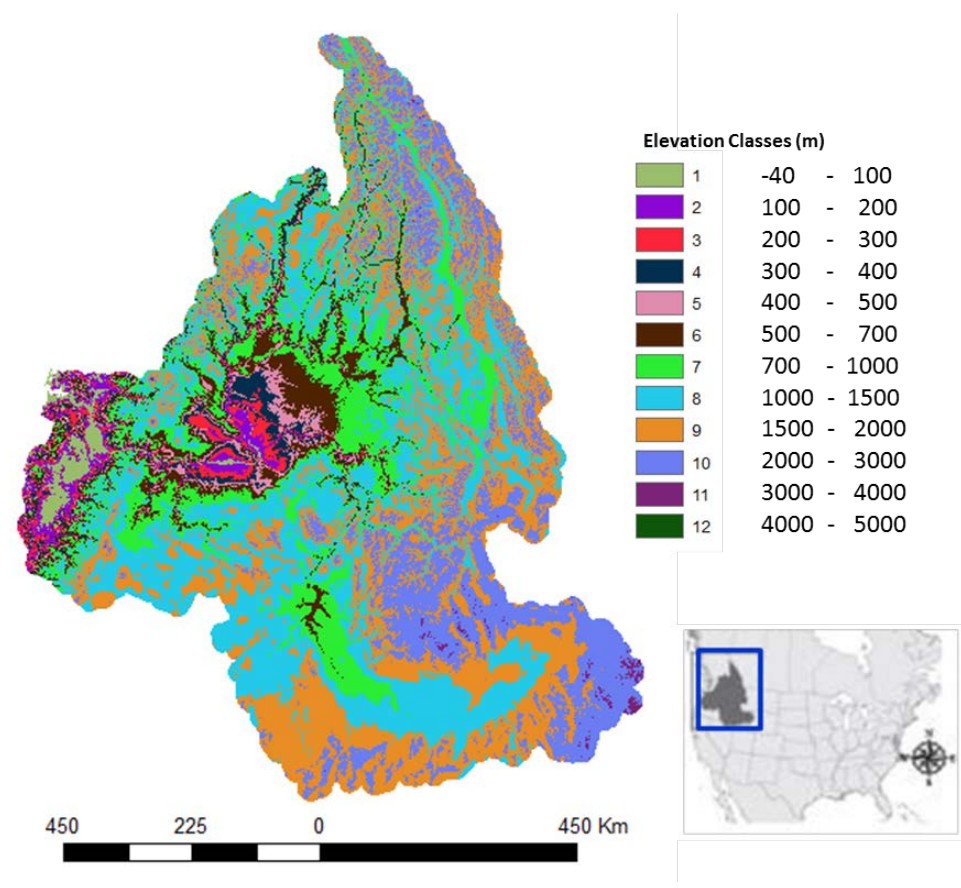

**Figure 2:** The study area classified into elevation bands used in the Global method, following the approach described in Leung and Ghan (1995; 1998).



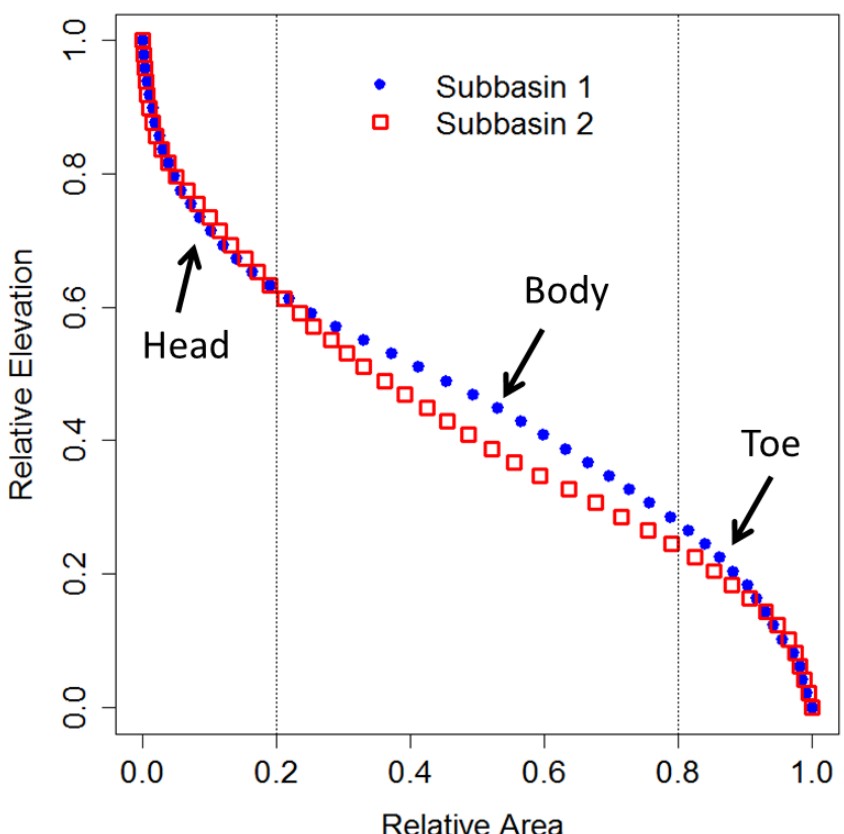

**Figure 3:** Hypsometric curves of two subbasins with extreme contrast of elevation variability discretized into three parts
following Willgoose and Hancock (1998) and Sinha-Roy (2002): the head, body and toe, as used in the Local method.





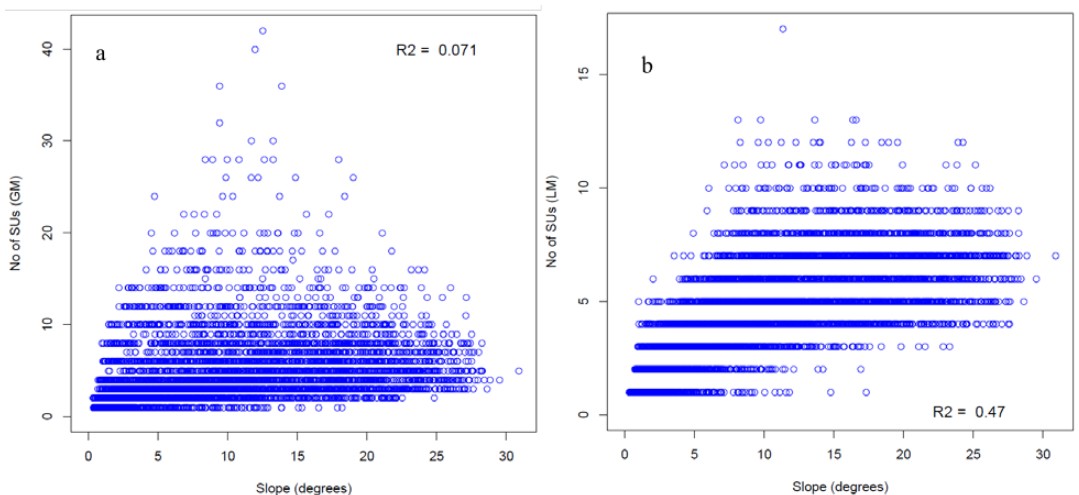

**Figure 4:** The number of elevation-based geo-located SUs plotted against the average topographic slope for each subbasin derived using the Global (a) and Local (b) methods with 1% area threshold.





**Figure 5:** Spatial patterns of the number of elevation-based geo-located SUs per subbasin derived using the Global (c) and Local (d) methods compared against the spatial pattern of the topographic slope (a) and elevation ranges of the subbasins in the study area.





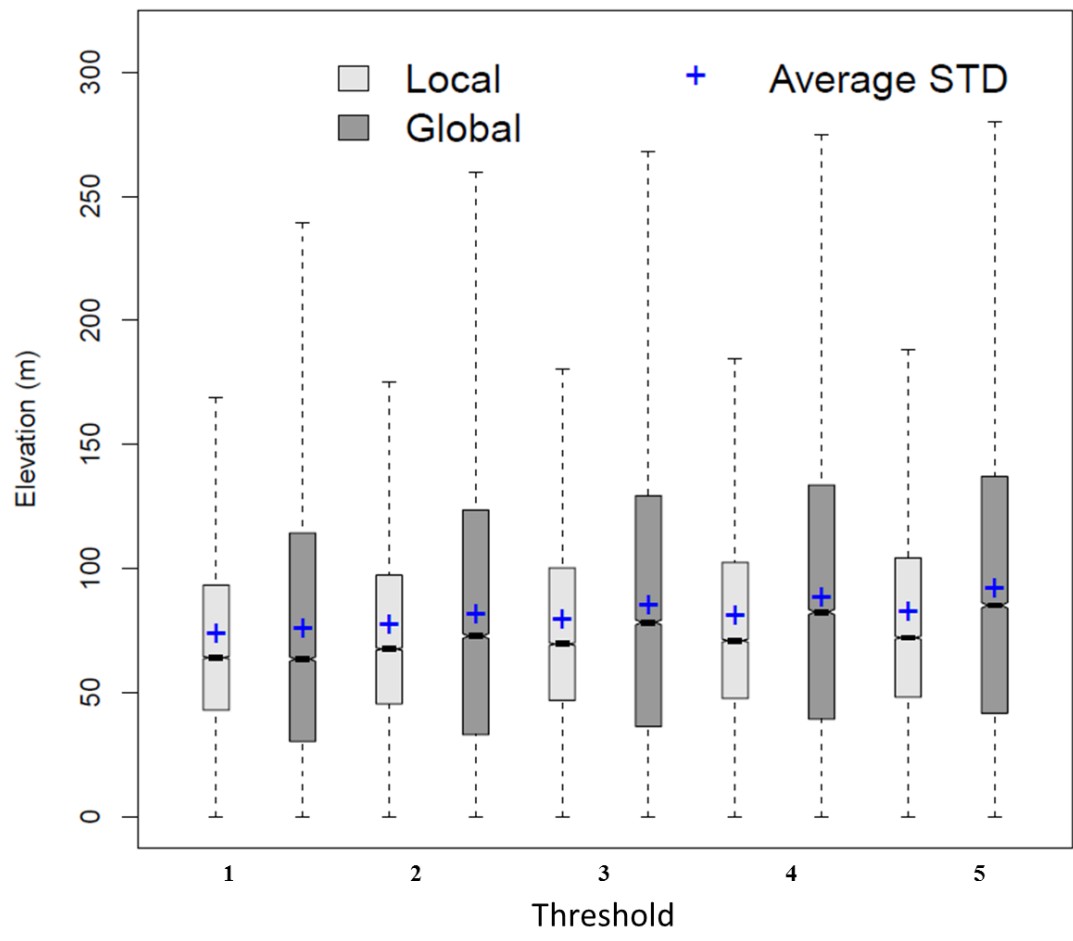


**Figure 6:** The standard deviation in elevation within the elevation-based geo-located SUs derived using different values of area threshold. On each box, the central mark (notch) is the median (q2), the edges of the boxplot are the 25th (q1) and 75th (q3) percentiles, and the whiskers extend to the most extreme data points (q3 + 1.5 x interquartile range (q3 – q1) and q1 – 1.5 x interquartile range (q3 – q1); outliers are not considered.






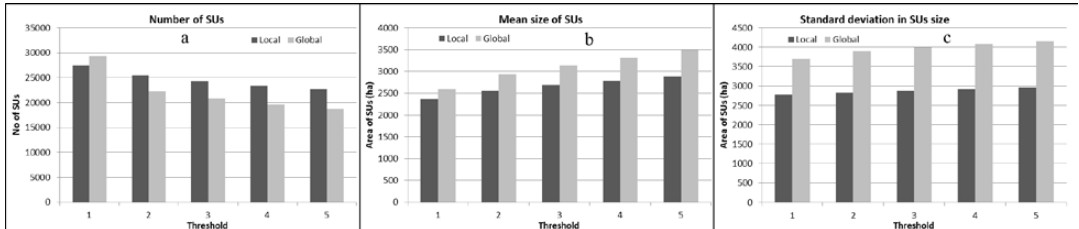

**Figure 7:** Sensitivity of the Global (grey) and Local (black) methods to different values of area threshold for the total number of SUs (a), average SUs size (b) and standard deviation in SU size (c) of the elevation-based geo-located SUs derived using
different values of area threshold.

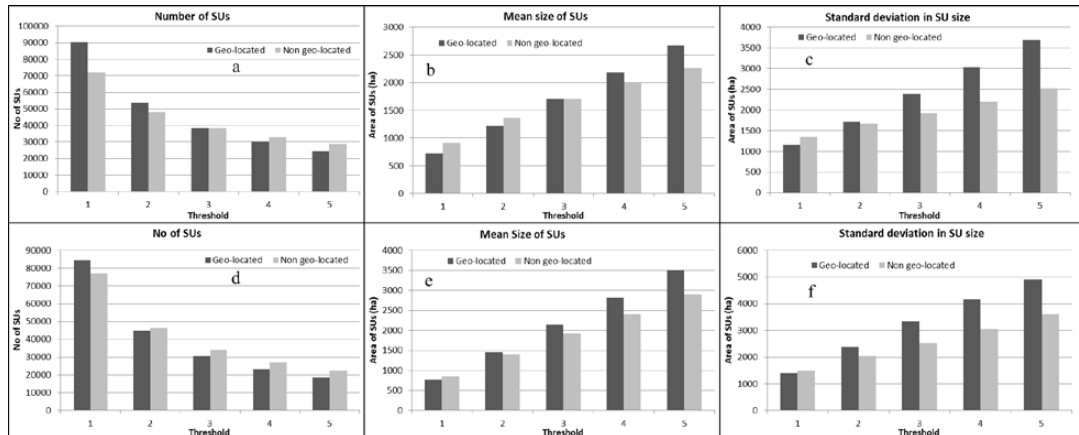

**Figure 8:** Comparison of the geo-located (black) versus non-geo-located (grey) SUs derived based on elevation, slope and aspect using the Global (a, b, and c) and Local (d, e, and f) methods, in terms of their sensitivity to different values of area threshold for
the total number of SUs, average SU size and standard deviation in SU size over the study domain.



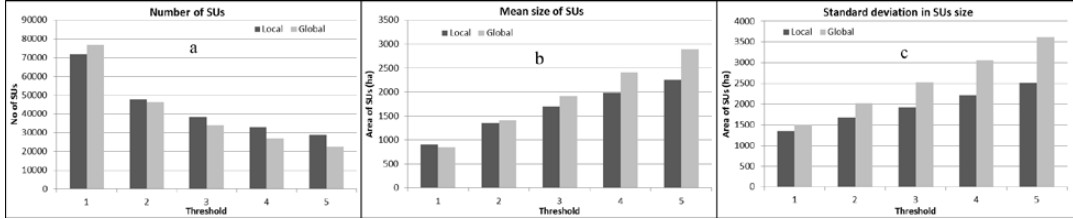

**Figure 9:** Comparison of the two methods (Global and Local) using non-geo-located SUs in terms of their sensitivity to different values of area threshold for the total number of SUs (a), average SU size (b) and standard deviation in SU size (c) over the study area. SUs are constructed based on elevation, slope, and aspect.





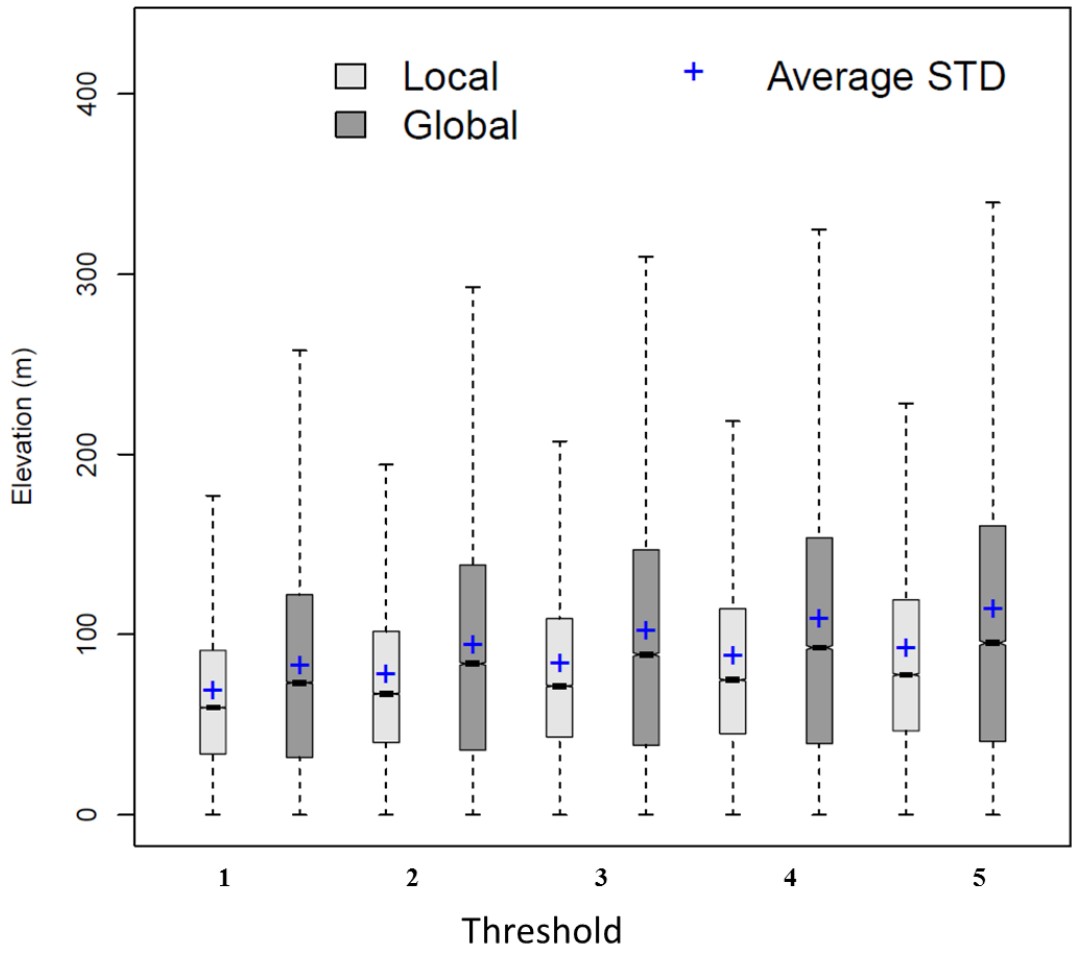

**Figure 10:** Similar to Figure 6, but for capability of the Global and Local methods to capture topographic heterogeneity based on the standard deviation in elevation within the non-geo-located SUs derived based on elevation, slope and aspect using different values of area threshold.






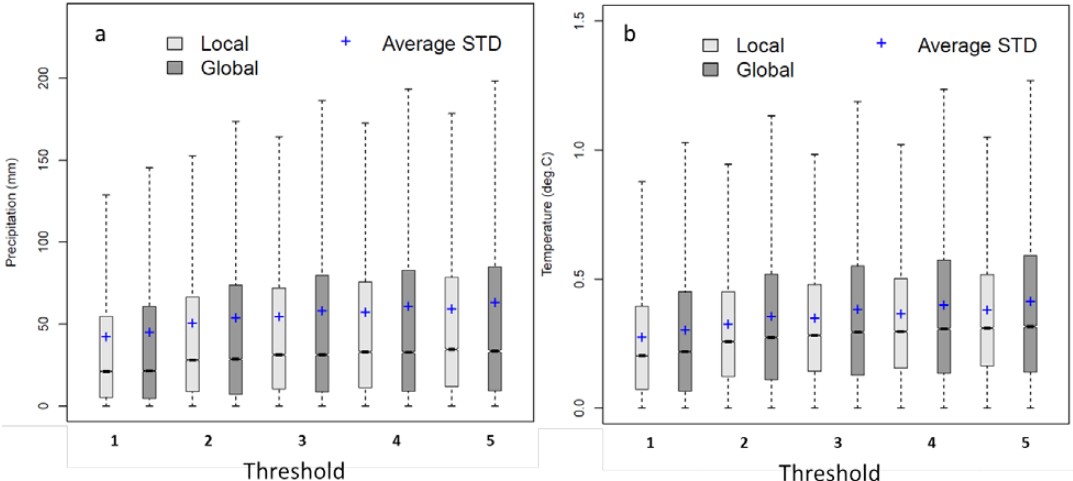

**Figure 11:** Similar to Figure 5, but for capability of the Global and Local methods to capture climatic variability based on standard deviation of the PRISM 30 year normal precipitation (a) and surface temperature (b) within the non-geo-located SUs derived based on elevation, slope and aspect across different values of area threshold.




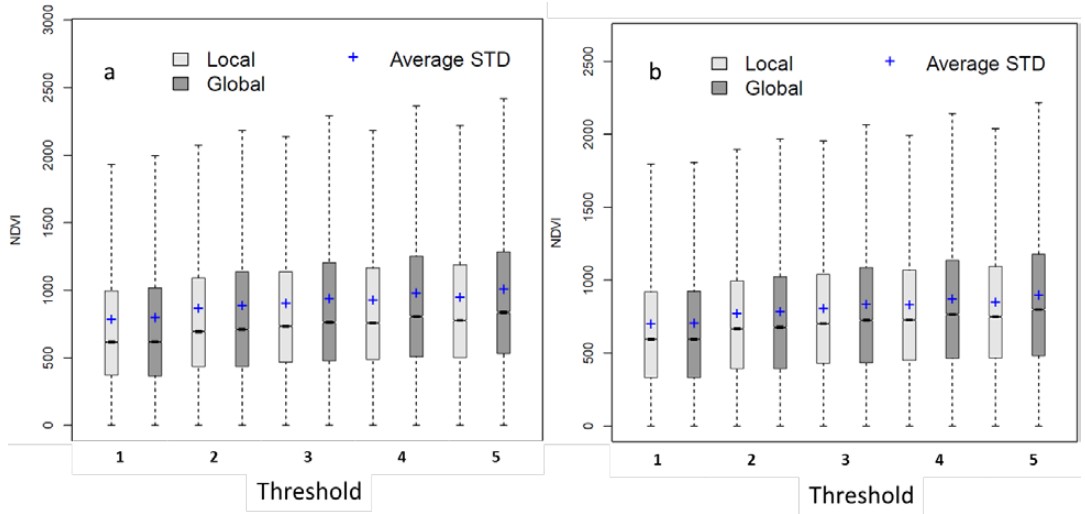

**Figure 12:** Similar to Figure 6, but for capability of the Global and Local methods to capture land cover variation based on standard deviation values of eMODIS NDVI during Spring (a) and Summer (b) within the non-geo-located SUs based on elevation, slope and aspect across different values of area threshold.


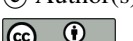



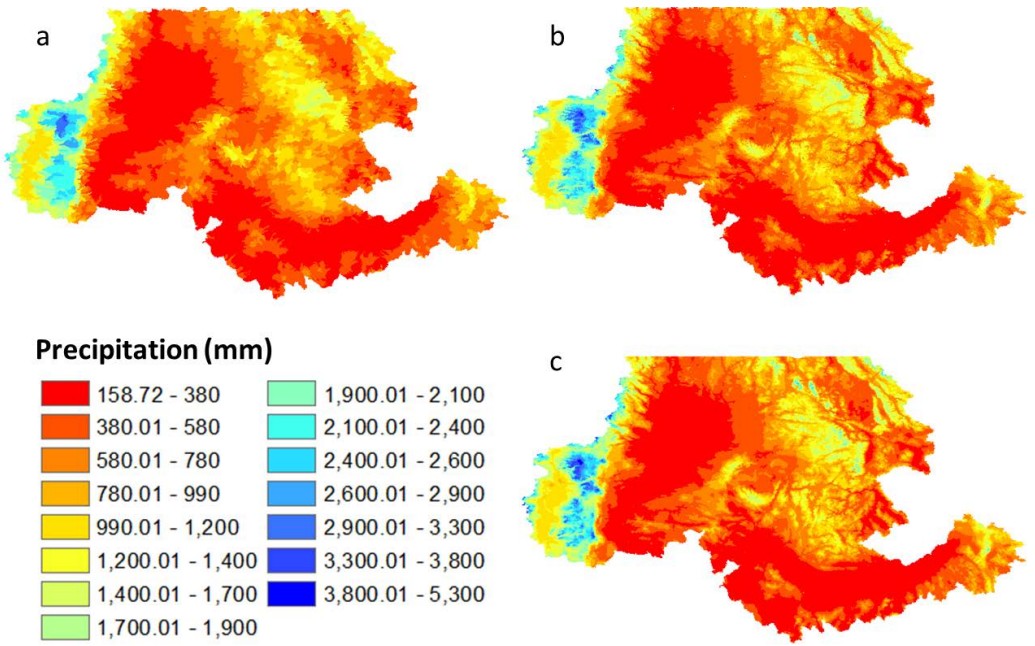

**Figure 13:** PRISM 30 year normal precipitation represented using the subbasins (a) and non-geo-located SUs based on elevation, slope and aspect from the Local method using 3% area threshold (b) compared to those of the original PRISM grids (c). The Canadian territory of the study area is not represented in the PRISM dataset.



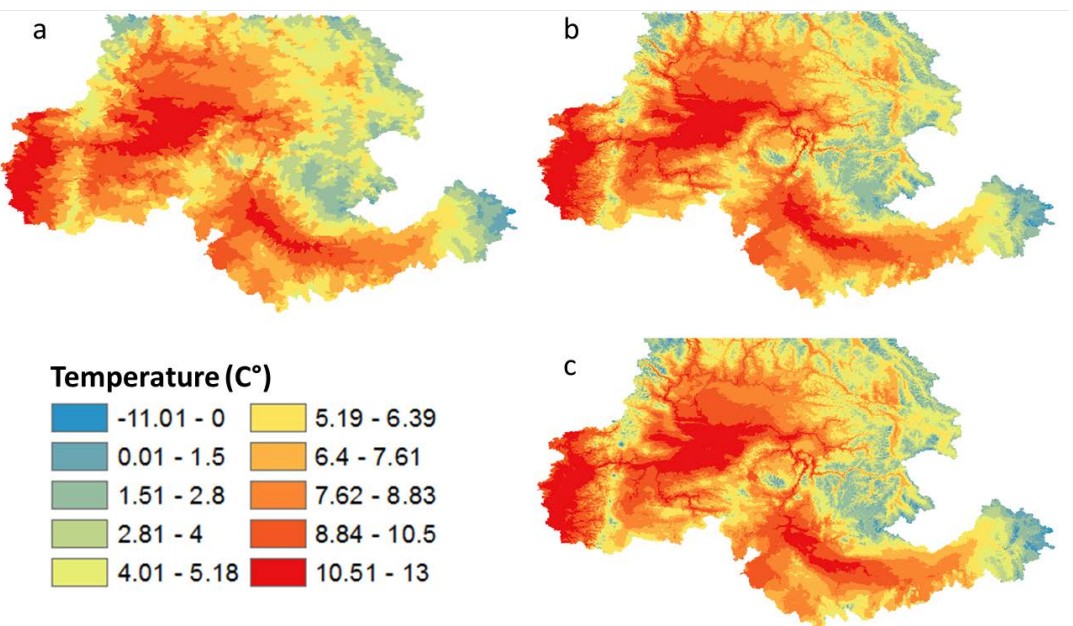

**Figure 14:** Same as Figure 13, but for temperature.
