# Peer review of "Exploring New Topography-based Subgrid Spatial Structures for Improving Land Surface Modeling"

_Geoscientific Model Development, 2016_

## Referee Comment (RC1) · Anonymous Referee #1 · 10 Aug 2016

This manuscript evaluated a new approach in representing the spatial heterogeneity of topography and pointed out that the representation based on a more flexible classification using hyposmetric analyses (local) and spatially non-contiguous (non-geo-located) subgrid structures is more robust. The manuscript is generally well written and I think it is ready to be published after the major points are answered.

Major comments:

1. In the atmospheric science field, the importance of land-surface processes to the evolution of temperature and moisture distribution in the atmospheric boundary layer is generally well recognized. The impact of spatial distribution of topography on the atmospheric motion and precipitation distribution, on the other hand, is a major topic in

the field (see, for example, the review paper by Houze 2012). With the new approach in representing subgrid structure of topography, can the atmospheric modeler benefit from the parameters used in the new approach for better representing subgrid scale land-topographic-precipitation processes?

2. The comparisons among various approaches (Global vs. Local, geo-located vs non-geolocated, etc.) in the paper are generally qualitatively rather than quantitative. For example, in line 240, I can understand the local method is better but I cannot understand how much better it is. Judging from the variability of the data, I can also argue that the two methods are roughly the same.

3. The purpose of using precipitation in the implications to representation of land surface processes is not clear to me. I think the goal of the new approach is to better capture the subgrid variability of the topography. Precipitation, on the other hand, is the overall results of land-atmosphere-topography interactions. Does that mean the atmospheric model should also have similar grid structure as the land surface model? In addition, I don't understand how the results in Figure 13b are better than Figure 13a.

Minor comments:

1. Line 61: Does the definition of subgrid affect the results? For example, the subgrid for the general circulation model grid size or the cloud resolving model grid size?

2. Line 83: Does the choice of study area affect the results?

3. Line 175: Can you be more specific on what area threshold means?

4. In line 230: "the spatial pattern of the number of SUs per subasin for the SUs from the Local method follows the topographic pattern in the study area better than those of the Global method". In Fig.5, it's difficult for me to recognize such point. Is it a result of coloring the number of SUs into 5 categories rather than 13 categories?

5. The Y-axis in Figures 7, 8, 9 and 12 is blurry and difficult to read.

Houze, R.A., 2012. Orographic effects on precipitating clouds. Reviews of Geophysics, 50(1).

---

## Referee Comment (RC2) · Anonymous Referee #2 · 8 Oct 2016

Current land surface models lack of addressing topographic information in their subgrid structures. In this study, the authors give two types of subgrid structures (geo-located and non-geo-located) over the topographically diverse Columbia River basin in the Northwestern United States using two topography-based methods (Local and Global) for watershed discretization, and the research topic is interesting and valuable. Generally, the methods are sound and have potential being used in land surface modeling. Therefore, the manuscript can be appcpted be published in the Journal Geosci. Model Dev. before some concerns given below have been addressed.

General comments:

1. The first concern is about the evaluation method. When comparing the Local and

Global methods, as well as the geo-located and non-geo-located structures, the authors tended to choose the options that are less sensitive to the values of area threshold because it can provide more robust subgrid structures for representing subgrid topographic heterogeneity. The sensitivity maybe be a key criterion to evaluate the subgrid structures. In reality, before simulation, we will set a certain area threshold based on the computational resources, the advantage of the discretization methods and subgrid structures should be evaluated under that certain area threshold. A more robust (or less sensitive) subgrid structure cannot ensure a better performance in a given area threshold. For example, in the Figure 8c and 8f, the standard deviations in subgrid structures size are lower for geo-located structure than non-geo-located with a 1% area threshold, and to the other thresholds, the situations are quite reverse.

2. Also, using the standard deviations in subgrid structure size to judge the performance of Local and Global methods is seens not reasonable (Figure 9). When applying Local method to discrete the subbasin, the size factor is implicitly included by dividing RA into several quasi-equal parts. However, in the Global method, the size factor is not considered. So it is natural that the standard deviation in subgrid structure size of Local method is lower than Global method. Moreover, the standard deviation in subgrid structure size is also not directly linked to the performance of each method or structure. It is better to define other criteria to judge the performance for each option or at least remove this unfair comparison from the manuscript.

3. When we compare two methods or two structures for subgrid scheme, the performance of each option under same computational task (number of subgrid structures) is expected, while an appropriate area threshold is pre-prescribed. In this study, there is no such comparison. Therefore, I think at least the author should find a threshold at which the two methods (or subgrid structures) share the same number of subgrid structures, and do the comparison (for the standard deviation in elevation, precipitation and temperature in this study) under this area threshold. From the Figure 8a, 8d and Figure 9a, I do believe such area threshold exists. However, to the Figure 8a, it is also

very curious that why the number of subgrid structures in non-geo-located structure can be more than it in the geo-located structure when area threshold is set to 4% and 5%. Intuitively, the number of SUs in non-geo-located structure should be always fewer because different subgrid structures (but with same elevation characteristics) in geo-located structure are combined to a same subgrid structures in the non-geo-located. The author should explain clearly about this abnormal effect.

Specific comments:

L144-146: The model code (Table1 and Table2) should be moved to supplementary material, and instead give a brief description about the procedure of what this code expresses in the manuscript.

L165-179: Please add another member that using Global method, geo-located structure and including the topographic slope in these comparisons to show that how the performance of Global method can be improved when the slope effect is considered.

L195: The title is not appropriate. No land surface processes were shown here, only precipitation and temperature.

L195-211: Some important information was missing here. How do the authors interpolate the precipitation (or temperature) from subbasin-level to subunits-level, or how can we get the Figure 14b from 14a? Much more detail information about proceeding method should be added.

L195-211: I think beside results from the option of Local method and non-geo-located structure, results from other options (Local method and geo-located structure, Global method and non-geo-located structure, Global method and geo-located structure) are also expected to be displayed in Table 3, Figure 13, and Figure 14.

L195-211: Please add spatial distribution for NDVI as it does to the precipitation in Figure 13. Also, the statistics about NDVI should be added in Table 3.

Figure 7, Figure 8 and Figure 9: These bar plots are not clearly displayed. Please

redraw these figures to make them discernible enough.

---

## Author Comment (AC1) · 6 Nov 2016

**Response to 1ˢᵗ Reviewer's comments**

We thank the reviewer for the insightful and constructive comments on our manuscript. The manuscript will be revised accordingly to address the comments. Specific responses to the reviewer's comments and revisions to be included in the manuscript are listed below.

**General Comment:**

*This manuscript evaluated a new approach in representing the spatial heterogeneity of topography and pointed out that the representation based on a more flexible classification using hypsometric analyses (local) and spatially non-contiguous (non-geo-located) subgrid structures is more robust. The manuscript is generally well written and I think it is ready to be published after the major points are answered.*

**Response:**

We thank the reviewer for the constructive comments on our manuscript. We will revise the manuscript accordingly to address all the reviewer's comments.

**Major comments:**

1. *In the atmospheric science field, the importance of land-surface processes to the evolution of temperature and moisture distribution in the atmospheric boundary layer is generally well recognized. The impact of spatial distribution of topography on the atmospheric motion and precipitation distribution, on the other hand, is a major topic in the field (see, for example, the review paper by Houze 2012). With the new approach in representing subgrid structure of topography, can the atmospheric modeler benefit from the paramters used in the new approach for better representing subgrid scale land-topographic-precipitation processes?*

**Response:**

As the reviewer pointed out, the impact of spatial distribution of topography on atmospheric motion and precipitation distribution is well recognized. Consequently, atmospheric modelers are actively working to enhance the capability of atmospheric models to capture the impact of topographic heterogeneity on precipitation distribution. This has been most commonly explored by increasing model resolution to better resolve topographic effects on mesoscale flow, cloud formation, and precipitation. Besides increasing model resolution, subgrid parameterizations have also been developed as a more computationally efficient approach. For example, Leung and Ghan [1995, 1998] developed a subgrid orographic precipitation parameterization to represent the impact of subgrid topography on airflow and precipitation for a discrete number of subgrid elevation classes defined using the non-geo-located Global method. They showed that this method produces realistic spatial distributions of precipitation and snow cover in mountainous areas. The subgrid structures presented in this study have been developed to improve the representation of land surface processes in land surface models. However, in a coupled modeling

framework, we envision the atmospheric model to adopt a subgrid structure similar to the non-geo-located subgrid units from the Local method, which as shown in the manuscript, capture more spatial heterogeneity of surface topography. The subgrid orographic precipitation scheme of Leung and Ghan will be included in the atmosphere model for coupling with the land surface model. The figure below is a global map of the number of subrid units per atmospheric grid developed as part of this effort described in the manuscript. We believe combining the subgrid orographic precipitation parameterization in the atmospheric model with the subgrid structure of the land model will provide the largest improvement for capturing subgrid variability of land surface processes. In the revised manuscript, we will clarify the motivation of our study and add some discussions about our plan for atmospheric modeling as elaborated above.

[Figure]

**Figure 1**: The spatial distribution of the number of subgrid unites per atmospheric grid (ne30np4) at global scale.

2. *The comparisons among various approaches (Global vs. Local, geo-located vs. non-geo-located, etc) in the paper are generally qualitatively rather than quantitative. For example, in line 240, I can understand the local method is better but I cannot understand how much better it is. Judging from the variability of the data, I can also argue that the two methods are roughly the same.*

**Response:**

We thank the reviewer for the constructive comment. To address this comment, we will include the following table in the revised manuscript and update the abstract, result and discussion, and summary and conclusions sections accordingly. Since the purpose of the subgrid method is to compare different subgrid spatial structures to better capture subgrid variability of topography, the most important metric is the reduction of standard deviation (STD) of subgrid topography

within each subgrid landunit (SU). As the table below shows, the Local Method generally reduces the STD of elevation by 17% - 19% compared to the Global Method. This quantifies the effectiveness of the Local Method over the Global Method for capturing subgrid variability of topography.

**Table 1**: Comparing the Local and Global methods in capturing topographic heterogeneity using non-geo-located SUs

| Average STD in elevation | | | |
|---|---|---|---|
| Area threshold (%) | Global Method | Local Method | Difference (%) |
| 1 | 80.81 | 66.82 | 17.30 |
| 2 | 92.10 | 75.77 | 17.73 |
| 3 | 100.03 | 81.60 | 18.43 |
| 4 | 106.55 | 86.20 | 19.10 |
| 5 | 112.14 | 90.48 | 19.32 |

3. *The purpose of using precipitation in the implications to representation of land surface processes is not clear to me. I think the goal of the new approach is to better capture the subgrid variability of the topography. Precipitation, on the other hand, is the overall results of land-atmosphere-topography interactions. Does that mean the atmospheric model should also have similar grid structure as the land surface model? In addition, I don't understand how the results in Figure 13b are better than Figure 13a.*

**Response:**

As discussed in our response to comment #1, our modeling objective is to implement a subgrid structure in both atmosphere and land models, together with a subgrid orographic precipitation scheme in the atmosphere model. Hence it is important to use existing precipitation and temperature datasets to evaluate the capability of the subgrid structures in capturing atmospheric forcing variability, as will be represented by the subgrid orographic precipitation scheme. For this purpose, Figure 11 compares the distribution of values of subgrid standard deviation in precipitation and temperature mapped from the high resolution PRISM datasets. Furthermore, in Figures 13 a, b, and c, we evaluate the spatial distribution of precipitation mapped to the subbasins and the non-geo-located subgrid structures from the Local method against the spatial distribution of precipitation from the original high resolution PRISM grid-based representation to determine whether the non-geo-located subgrid structures are able to improve representation of precipitation as compared to the subbasin-based representation. The subbasin-based representation used in this comparison comes from our previous studies (Tesfa et al., 2014a, 2014b), which evaluated the benefits of land surface modeling using subbasin-based approach against the standard regular grid-based land surface modeling approach, where significant

advantages in simulations of hydrologic fluxes and streamflow were demonstrated by the subbasin-based approach. However, we agree with the reviewer's comment that without a closer look at the mountainous areas (e.g., in Figure 13), it is not easy to visualize the improvement resulting from the new subgrid structure. Nevertheless with close inspection, it is possible to see the improvement, where the map from the new subgrid structure is more similar to the original PRISM grid representation than that of the Subbasin-based map. In addition, since statistical metrics are generally more informative than spatial maps, we have included statistical metrics in Table 3.

The atmospheric model will use non-geo-located subgrid structures derived based on the atmospheric grid as shown in Figure 1 in our response to comment #1.

In the revised manuscript, we will add more clarifications on how the PRISM climatic data are used to evaluate the subgrid structures.

**Minor comments:**

1. *Line 61: Does the definition of subgrid affect the results? For example, the subgrid for general circulation model grid size or the cloud resolving model grid size?*

**Response:**

The approaches described in the manuscript have been exclusively designed for subbasin/watershed based representation. The Local method, for example, utilizes a geomorphologic concept (hypsometric analysis) watershed analysis to derive the subgrid structures capturing the topographic pattern of the study domain. Application of the hypsometric analysis over the general circulation grid is not expected to yield the same behavior as that of the subbasin/watershed grid. However, it is possible to device a variant of the Local method capable to derive non-geo-located subgrid structures for the general circulation model grid size similar to those of the Local method described in the manuscript. For example, as part of our study to improve representation of orographic precipitation in the atmospheric model, we were able to replace the hypsometric analysis (Figure 3 in the manuscript) in the Local method by area-elevation profile curves and discretized each atmospheric grid (ne30np4) into surface elevation-based non-geo-located subgrid structures capable of capturing topographic heterogeneity (please see Figure 1).

2. *Line 83: Does the choice of study area affect results?*

**Response:**

Yes, the choice of the study area may affect the results. For example, the Local method has been designed to derive the subgrid structures in a way that minimizes computational demand by discretizing mountainous areas into more subgrid units and flat areas into a smaller number of subgrid units. Thus, the advantages of the new subgrid structures from the Local method are

expected to be more pronounced when applied over topographically heterogeneous/mountainous areas as opposed to areas characterized by homogenous/flat topography.

In the revised manuscript, we will state that the advantages of the non-geo-located subgrid structures from the Local method are expected to be more pronounced over areas characterized by heterogeneous topography.

*3. Line 175: Can you be more specific on what area threshold means?*

**Response:**

An area threshold is a value calculated as a percentage of the area of each subbasin to be used as a criterion for identifying smaller subgrid units that should be merged to their neighboring larger subgrid units to enable discretization of each subbasin into a reasonable number of subgrid structures. As it has been demonstrated in our response to the 2$^{nd}$ reviewer's comment #3, both methods (Global and Local) initially discretize each subbasin into many subgrid units. To divide the subbasin into a reasonable number of subgrid units, the normalized area of each subgrid unit, expressed as a percentage of the area of the subbasin, is calculated and compared with the value of the area threshold. All subgrid units with normalized areas smaller than the threshold are then merged to the neighboring larger subgrid units.

In the revised manuscript, more clarification of the area threshold value will be included.

*4. In line 230: "the spatial pattern of the number of SUs per subbasin for the SUs from the Local method follows the topographic pattern in the study area better than those of the Global method". In Fig. 5, it's difficult for me to recognize such point. Is it a result of coloring the number of SUs into 5 categories rather than 13 categories?*

**Response:**

We thank the reviewer for the comment. However, we think it is quite obvious from Fig. 5c and Fig. 5d that the numbers of SUs used in the two methods are very different. For example, there is a larger east-west gradient in the number of SUs in the mid- and upper-basin in Fig. 5c compared to Fig. 5d. Also in the western basin, the variations of the number of SUs in Fig. 5d correspond much better to the spatial variations of topography than that shown in Fig. 5c. To quantify the correspondence between the pattern of surface topography and the pattern of the number of SUs, we will present correlation coefficients between the two for the Global and Local method in the revised manuscript. We also want to note that the statement in the manuscript that the spatial pattern of the number of SUs per subbasin for the SUs from the Local method follows the topographic pattern in the study domain better than those of the Global method is also supported by the results shown in Figure 4. In Figure 4, the number of the subgrid units per subbasin from the Local method correlated with the values of the average slope of the subbasins much better than those of the Global method.

In the revised manuscript, we will include similar clarifications.

*5. The Y-axis in Figures 7, 8, 9 and 12 is blurry and difficult to read.*

**Response plan:**

We thank the reviewer for the comment. In the revised manuscript, Figures 7, 8, 9, and 12 will be updated to improve the quality of the figures.

**References:**

Leung, L. R. and Ghan, S. J.: A subgrid parameterization of orographic precipitation, Theoretical and Applied Climatology, 52, 95-118, 1995.

Leung, L. R. and Ghan, S. J.: Parameterizing Subgrid Orographic Precipitation and Surface Cover in Climate Models, Monthly Weather Review, 126, 3271-3291, 1998.

Tesfa, T. K., Li, H. Y., Leung, L. R., Huang, M., Ke, Y., Sun, Y., and Liu, Y.: A subbasin-based framework to represent land surface processes in an Earth system model, Geosci. Model Dev., 7, 947-963, 2014a.

Tesfa, T. K., Ruby Leung, L., Huang, M., Li, H.-Y., Voisin, N., and Wigmosta, M. S.: Scalability of grid- and subbasin-based land surface modeling approaches for hydrologic simulations, Journal of Geophysical Research: Atmospheres, 119, 3166-3184, 2014b.

---

## Author Comment (AC2) · 6 Nov 2016

**Response to 2nd Reviewer's comments**

We would like to thank the reviewer for the constructive comments on our manuscript. We will revise the manuscript accordingly to address all the comments. Detailed responses to the reviewer's comments and revisions planned are listed below.

**Overall Comment:**

*Current land surface models lack of addressing topographic information in their subgrid structures. In this study, the authors give two types of subgrid structures (geo-located and non-geo-located) over the topographically diverse Columbia River basin in the Northwest Untied States using two topography-based methods (Local and Global) for watershed discretization, and the research topic is interesting and valuable. Generally, the methods are sound and have potential being used in land surface modeling. Therefore, the manuscript can be accepted be published in the Journal Geosci. Model Dev. before some concerns given below have been addressed.*

**Response:**

We thank the reviewer for the constructive comments. The manuscript will be revised accordingly to address all the comments.

**General comments:**

1. *The first concern is about the evaluation method. When comparing the Local and Global methods, as well as the geo-located and non-geo-located structures, the authors tended to choose the options that are less sensitive to the values of area threshold because it can provide more robust subgrid structures for representing subgrid topographic heterogeneity. The sensitivity may be a key criterion to evaluate the subgrid structures. In reality, before simulation, we will set a certain area threshold based on the computational resources, the advantage of the discretization methods and subgrid structures should be evaluated under that certain area threshold. A more robust (or less sensitive) subgrid structure cannot ensure a better performance in a given area threshold. For example, in the Figure 8c and 8f, the standard deviations in subgrid structures size are lower for geo-located structure than non-geo-located with 1% area threshold, and to the other thresholds, the situations are quite reverse.*

**Response:**

We thank the reviewer for the comments. We agree with the reviewer that a less sensitive subgrid structure to threshold does not ensure better performance in a given area threshold. However, the purpose of the comparison in Figure 8 is only to show the sensitivity of the two types of subgrid structures (geo-locate and non-geo-located) to the values of area threshold; it is not intended to compare the performance of the two methods. The performance of the subgrid

structures in capturing topographic heterogeneity and climate and vegetation variabilities are evaluated using the results shown in Figures 4, 5, 6, 10, 11 and 12.

Regarding the area threshold, the geo-located subgrid units discretize the study domain into geographically contiguous units, while the non-geo-located subgrid units divide the area into geographically non-contiguous subgrid units. This means, before applying area threshold values to merge the units with values of percentage area less than the threshold, for a given study domain, the following relationship always holds true:

$$numGeo >= numNonGeo \qquad\qquad\qquad (1)$$

where, numGeo and numNonGeo are the number of geo-located and non-geo-located subgrid units, respectively. However, this may not be true after area threshold values are applied to merge the smaller subgrid units with the neighboring larger units. Compared to the geo-located subgrid units, the non-geo-located subgrid units tend to have a smaller number of units with size less than the area threshold values because many small subgrid units that are not spatially contiguous but with the same topographic characteristics are grouped into one subgrid unit and their combined area tends to become larger than the area threshold value. Thus, for a given area threshold, more subgrid units are merged with the neighboring units in the geo-located type than those of the non-geo-located subgrid units. Therefore, as the value of area threshold increases, the number of geo-located subgrid units can be less than the number of non-geo-located subgrid units for a given study domain. Please see the example watersheds provided in our response to comment #3 for more clarification.

2.  *Also, using the standard deviations in subgrid structure size to judge the performance of the Local and Global methods seems not reasonable (Figure 9). When applying Local method to discrete the subbasin, the size factor is implicitly included by dividing RA into several quasi-equal parts. However in the Global method, the size factor is not considered. So it is natural that the standard deviation in subgrid structure size of Local method is lower than Global method. Moreover, the standard deviation in subgrid structure size is also not directly linked to the performance of each method or structure. It is better to define other criteria to judge the performance for each option or at least remove this unfair comparison from the manuscript.*

**Response:**

We thank the reviewer for the comment. We agree that the performance of the methods (Global and Local) is not linked with the standard deviation in subgrid structure size. Similar to Figure 8, the results in Figure 9 only aim to compare the sensitivity of the non-geo-located subgrid units to the values of area threshold when derived using the Global and Local methods. Performance of the two methods (Global and Local) in capturing topographic, climatic and vegetation heterogeneity are evaluated in Figures 10, 11 and 12. The results in Figure 9 clearly show that the non-geo-located subgrid structures from the Local method are less sensitive to the values of

area threshold as compared to those of the Global method. We also agree with the reviewer's comment that, taking advantage of the hypsometric analysis, the Local method divides the subbasin into quasi-equal parts resulting in non-geo-located subgrid structures that are less sensitive to the values of area threshold.

In the revised manuscript, we will clarify the purpose of the results compared in Figures 8 and 9 as elaborated above.

3. *When we compare two methods or two structures for subgrid for subgrid scheme, the performance of each option under same computational task (number of subgrid structures) is expected, while an appropriate area threshold is pre-prescribed. In this study, there is no such comparison. Therefore, I think at least the author should find a threshold at which the two methods (or subgrid structures) share the same number of subgrid structures, and do the comparison (for the standard deviation in elevation, precipitation and temperature in this study) under this area threshold. From the Figure 8a, 8d and Figure 9a, I do believe such area threshold exists. However, to the Figure 8a, it is also very curious that why the number of subgrid stuructures in non-geo-located structures can be more than it in the geo-located structures when area threshold is set to 4% and 5%. Intuitively, the number of SUs in non-geo-located structure should be always fewer because different subgrid structures (but the same elevation characteristics) in geo-located structure are combined to a same subgrid structures in the non-geo-located. The author should explain clearly about this abnormal effect.*

**Response:**

We thank the reviewer for the constructive comment. The comment can be divided into two parts. First, the reviewer suggests finding a threshold value that gives the same number of subgrid structures in both methods to repeat the comparison of the performance of the two methods at that threshold. Although this is doable, we think that the comparison is already in the manuscript. Figure 9a shows that the area threshold value resulting in the same number of non-geo-located subgrid units from the two methods (Global and Local) lies between 1% and 2%. Figure 10, on the other hand compares the non-geo-located subgrid units from the two methods across different values of area threshold (1%, 2%, 3%, 4%, and 5%) for their capability to capture topographic heterogeneity. The result for the area threshold value resulting in the same number of subgrid units in both methods is implicitly included in Figure 10 between area threshold values of 1% and 2%. Therefore, we think the first part of the reviewer's comment has been already addressed in the manuscript and we will add some discussions for such a comparison.

The second part of the comment is the concern why the number of non-geo-located becomes greater than that of the geo-located at thresholds of 4 and 5 percent. Since this comment is similar to comment #1, the reasons are described in our response to comment #1. However, for

more clarification, the condition is demonstrated using one example subbasin from the study domain, as described below

To clarify why the number of non-geo-located subgrid units becomes greater than the number of the geo-located units as we increase the values of area threshold, we derived geo-located and non-geo-located subgrid units for one of the subbasins selected to represent extreme classes of elevation ranges in Figure 1 in the manuscript. The figure below shows the subbasin discretized into geo-located (left) and non-geo-located (right) subgrid units using area threshold of 1%. The colors represent the identification numbers of the units; thus, areas of the same color in the non-geo-located subgrid units belong to the same unit. In this example, the Global method was applied for six values of area threshold (1%, 2%, 3%, 4%, 5%, and 6%) to discretize the subbasin into both geo-located and non-geo-located SUs. The numbers of subgrid units are then compared before and after applying values of area threshold. Note that area threshold values are used to merge all units less than the threshold to the neighboring larger units.

As shown in the results below, the number of geo-located subgrid units is greater than those of the non-geo-located units before applying the area threshold values. However, after using the area threshold values to merge all the small subgrid units with their neighboring larger units, the number of the geo-located subgrid units decreases with increasing threshold faster than those of the non-geo-located subgrid units. As explained in our response to comment #1, geo-located subgrid type tends to have more subgrid units with size smaller than the threshold than those of the non-geo-located type, resulting in more subgrid units being merged as the threshold values increase.

Geo-located SUs (1%)     Non- geo-located SUs (1%)

[Figure]

| Threshold | No of initial SUs | | No of final SUs | |
|---|---|---|---|---|
| | Geo-located | Non-geo-located | Geo-located | Non-geo_located |
| 1 | 808 | 699 | 23 | 13 |
| 2 | 808 | 699 | 11 | 8 |
| 3 | 808 | 699 | 7 | 8 |
| 4 | 808 | 699 | 6 | 6 |
| 5 | 808 | 699 | 5 | 5 |
| 6 | 808 | 699 | 3 | 4 |

**Figure 1**: Comparison of the number of geo-located and non-geo-located subgrid units derived using the Global method across different values of area threshold.

**Specific comments:**

1. *L144 – 146: the model code (Table1 and Table2) should be moved to the supplementary material, and instead give a brief description about the procedure of what this code expresses in the manuscript.*

**Response:**

We thank the reviewer for the suggestion. In the revised manuscript, Tables 1 & 2 will be moved to the supplementary material and the manuscript will be updated with the description of the procedures used in the code.

2. *L165 – 179: Please add another member that using Global method, geo-located structure and including the topographic slope in these comparisons to show that how the performance of Global method can be improved when the slope effect is considered.*

**Response:**

We thank the reviewer for the constructive suggestion. Subsection 3.5 of the manuscript aims to evaluate the two methods (Global and Local) using geo-located subgrid structures derived based on elevation classification only. We agree with the reviewer on the importance of showing how the performance of the Global method can be improved when slope effect is considered. However, to keep the story smooth and the manuscript concise, we think such comparison should be placed in the supplementary materials.

In the revised manuscript, we will include the figure shown below in the supplementary material, which compares the number of geo-located subgrid units per subbasin derived using the Global method based on combination of topographic elevation and slope against the average slope of the subbasins to show how the Global method performs when effect of slope is considered. The result and discussion section will also be updated to discuss the comparison.

[Figure]

**Figure 2**: Number of geo-located subgrid units per subbasin from the Global method based on the combination of topographic elevation and slope at area threshold value of 1% compared against values of average slope of the subbasins

3.  *L195: The title is not appropriate. NO Land surface processes were shown here, only precipitation and temperature.*

**Response:**

We thank the reviewer for the comment. The title will be changed to "Exploring New Topography-based Subgrid Spatial Structures to improve representation of land surface processes in land surface models".

4.  *L195 – 211: Some important information missing here. How do the authors interpolate the precipitation (or temperature) from subbasin-level to subunits-level, or how can we get the Figure 14b from 14a? Much more detail information about proceeding method should be added.*

**Response:**

The PRISM climate dataset has a spatial resolution of 800m. Both precipitation and temperature values for each subbasin/subgrid unit are computed as the average of the corresponding values from all the source PRISM grids that intersect with the subbasin/subgrid unit.

In the revised manuscript, more detailed description of the mapping approach will be included.

5. *L195-211: I think beside results from the option of Local method and non-geo-located structures, results from other options (Local method and geo-located stuructures, Global method and non-geo-located structure, Global method and geo-located strucutres) are also expected to be displayed in Table 3, Figure 13, and Figure 14.*

**Response:**

We thank the reviewer for the comment. The manuscript has been designed to evaluate the two methods (Global and Local) and two types of subgrid structures (geo-located and non-geo-located) in a more orderly manner. For this purpose, first the two methods (Global and Local) are compared using the geo-located subgrid units for their capability to capture topographic heterogeneity and to demonstrate the benefits of using hypsometric analysis in the Local method in capturing variability of topographic slope implicitly (Figures 4, 5, 6 , and 7). The geo-located subgrid structures are then compared against the non-geo-located (Figure 8). The two methods are then compared for their sensitivity to the values of area threshold (Figure 9). Furthermore, the two methods (Global and Local) are compared using the non-geo-located subgrid structures for their capability to capture topographic, climate and vegetation heterogeneity (Figures 10, 11, and 12). Finally, the method and subgrid structure type with more pronounced advantages are compared against the subbasin-based representation to evaluate potential improvements of the new subgrid structures (Figures 13 and 14). We think including the results from Local method and geo-located, Global method and non-geo-located subgrid structure, and Global method and geo-located subgrid structure would make the manuscript too long. To address the reviewer's comments without making the manuscript too long, in the revised manuscript, table 3 will be extended with results from Global and non-geo-located subgrid structures, and maps similar to Figures 13 and 14 for the non-geo-located subgrid structures from the Global method will be included in the supplementary materials.

6. *L195 – 211: Please add spatial distribution for NDVI as it does to the precipitation in Figure 13. Also, the statistics about NDVI should be added in Table 3.*

**Response:**

We thank the reviewer for the comment. Since statistical metrics are generally more informative than maps, to address the reviewer's comments without making the paper too long, we will include a separate table (table 4) in the revised manuscript with the following comparisons.

- Results from Global and non-geo-located subgrid structures.
- Results from the Local method and non-geo-located subgrid structures.
- Results from the subbasin-based representation.

7. *Figure 7, Figure 8 and Figure 9: These bar plots are not clearly displayed. Please redraw figures to make them discernible enough.*

**Response:**

We thank the reviewer for the comment. In the revised manuscript, the figures will be updated as recommended to improve the quality of the figures.